# Exploring the influence of social media use motivations on acculturation orientations and psychological adaptation among Chinese students in Malaysia

Wenwen Zhao🆔*

School of Literature and Journalism, Guangdong Ocean University, China

\* zhaowenwen0920@gmail.com

## Abstract

Social media has become an essential part of international students' daily lives. As the world becomes increasingly globalized and communication and transportation systems rapidly develop, the number of international students is growing remarkably. Consequently, the influence of social media on the psychological adaptation of international students has become a matter of concern. However, the relationship between these two factors remains inconclusive. Moreover, there is a scarcity of studies investigating the influence of social media use motivations on the psychological adaptation of international students. Previous research has evidenced that the relationship is indirect, but limited studies have identified psychological mechanisms such as acculturation orientations. Therefore, this study aims to identify the mediating role of acculturation orientations (Malaysian and Chinese cultural orientations) in the relationship between social media use motivations (information seeking, social, and entertainment motivations) and the psychological adaptation of Chinese students in Malaysia. An online survey was administered, employing the convenience sampling technique to collect data from a sample of Chinese international students in Malaysia. A total of 438 questionnaires were analysed by using SPSS and AMOS software. The results demonstrated that three types of use motivations can help Chinese students improve their psychological adaptation in Malaysia, while entertainment motivations have the greatest influence. Likewise, acculturation orientations also benefited psychological adaptation, with mainstream orientations having a stronger contribution. Furthermore, this study reported an indirect mechanism whereby information seeking and social motivations influenced psychological adaptation through Malaysian and Chinese cultural orientations. The findings suggested that Chinese students can use social media purposefully to obtain a better psychological adaptation in Malaysia. Besides, social media designers can enhance the functions and services of social media to provide more support for international students. Relevant departments and

**Data availability statement:** The data are available from figshare at DOI: 10.6084/m9.figshare.28379111 (https://doi.org/10.6084/m9.figshare.28379111)."

**Funding:** This work was supported by Humanities and Social Science Project of Guangdong Ocean University: Study on social media use and acculturation of sojourn (Grant No. 030301142302). The funders had no role in study design, data collection and analysis, decision to publish, or preparation of the manuscript.

**Competing interests:** The authors have declared that no competing interests exist.

educators in Malaysia should appreciate the role of social media and acculturation orientations in helping international students have a better overseas experience and enhance Malaysia's competitiveness in the international education market.

## Introduction

With the escalating speed, intensity, frequency, and number of human mobility increases, the international student population is experiencing rapid growth [1]. As stated by the United Nations Educational, Scientific and Cultural Organization (UNESCO), the number of international students globally surpassed 6.4 million in 2021, up from 2.1 million in 2000 [2]. International students entering a new environment are accompanied by various challenges, such as homesickness, language barriers, unfamiliar cultural environments, lack of social support, and academic pressures [3,4], which may negatively influence their psychological adaptation in the host country [5]. Poor psychological adaptation may lead to severe consequences, such as dropping out of their courses [6]. The high dropout rate of international students has become a significant problem for international education development [6].

In an unfamiliar environment, international students are more likely to use social media to connect with friends and family, build new social networks, obtain necessary information, learn about the host culture, and thus better adapt to the host country [7]. Consequently, international students use social media more actively [8]. Using social media to stay in touch with family, friends, and others has become an essential aspect of international students' daily lives [9].

However, the side effects of social media use have been demonstrated [10]. Excessive engagement with social media by international students can hinder their integration into the host culture [10,11]. Moreover, social media usage among international students can lead to distractions, disrupted sleep, and peer pressure [12]. Additionally, the constant exposure to information on social media can result in information overload, further impairing their ability to concentrate on their academic pursuits [13]. Furthermore, one of the most concerning issues in social media research is its influence on users' mental health [14]. Therefore, considering the increasing popularity of social media in international students' daily lives and the substantial increase in the number of international students, it is crucial to acknowledge social media and its influence on their psychological adaptation.

Notably, scholars have different conclusions about whether social media benefits or harms psychological adaptation. Social media is not inherently positive or negative; rather, it depends on how users use it [15]. Therefore, the pressing issue lies in addressing how to effectively utilize social media and maximize its positive influence on the psychological adaptation of international students. Previous studies have underscored the influence of use motivations on the connection between social media and psychological outcomes [16,17]. It is essential to give more attention to understanding the motivations behind the use of social media and its influence on the psychological adaptation of international students [18].

Besides, scholars considered that the existence of mediators between social media use and psychological adaptation [16,19]. However, few studies examine how social media influences adaptation [19,20]. While some previous studies have suggested potential mechanisms like social support [16,21], the exploration of significant psychological mechanisms, such as acculturation remains relatively understudied. Besides, research on the relationship between social media use, acculturation, and psychological outcomes of international students is limited [22].

Malaysia has become a popular destination for Chinese students to study abroad [23]. As reported by the Ministry of Higher Education (MoHE) Malaysia, the year 2021 saw more than 28,000 Chinese students pursuing higher education in Malaysian institutions, constituting 29.4% of the total international student body in the country. Chinese students have become Malaysia's most prominent international student group [24]. However, the majority of research on Chinese international students' adaptation have been conducted in Western context. Moreover, limited research has focused on the interplay relationship between social media, acculturation orientations, and the psychological adaptation of Chinese students in Malaysia. Therefore, it is of great significance for this study to focus on this issue.

## Theory and literature review and research hypothesis

### Acculturation theory

Acculturation originated from anthropology and was first proposed by Redfield et al. (1936), who suggested that "acculturation comprehends those phenomena which result when groups of individuals having different cultures come into continuous first-hand contact with subsequent changes in the original cultural patterns of either or both groups" [25, p. 149]. Nowadays, globalization, brought about by rapid technological developments such as media and transport, has produced new forms of acculturation [26]. For example, the previous emphasis on "continuous first-hand contact" may not be necessary. For example, international students can be exposed to the host culture directly offline or indirectly through the media.

Berry (2003) proposed a framework for understanding the acculturation process, in which the contact between two cultures would produce changes at the group (cultural) level (e.g., social and cultural systems) and individual (psychological) level (e.g., behavioral shifts and acculturative stress). These behavioral shifts are often observed in an individual's speech patterns, food preferences, and cultural identity [27]. Thus, acculturation is "the dual process of cultural and psychological change that takes place as a result of contact between two or more cultural groups and their individual members" [28], p. 698]. When two cultures interact, the primary concern lies in how individuals view the host culture and their ethnic culture [29]. In summary, acculturation theory provides insights into how individuals adjust their cultural behaviors, values, and identities to cope with new cultural contexts and reflects "the process in which one cultural group accepts the beliefs and behaviors of another cultural group" [30, p. 110].

Berry (1980) suggested a bi-dimensional framework, which holds that ethnic orientations and mainstream orientations are independent of each other [31]. Recent studies have supported this opinion and indicated that individuals who maintain mainstream orientations do not necessarily reject their ethnic culture [22,32]. A considerable number of international students are inclined to return to their home country after residing in the host country for a period, thereby making the bi-dimensional approach a more relevant theoretical framework [10]. In the current study, mainstream culture refers to Malaysian culture, and ethnic culture refers to Chinese culture.

Besides, acculturation can be viewed as a process of linear causality, in which some predictors affect acculturation orientations and thus influence adaptation consequences [33]. Previous studies have shown that social media is a crucial predictor of acculturation [17,34]. Furthermore, the changes experienced during cultural contact ultimately influence psychological adaptation [35], which is the central outcome of acculturation [36]. Social media was a beneficial tool to influence the acculturation orientations and psychological well-being of Chinese migrants in the host country [22]. Based on acculturation theory and previous literature, this study holds that social media use motivations influence acculturation orientations and thus affect psychological adaptation.

## Social media use motivations

Social media use motivations refer to individuals choosing specific media purposely in a particular environment to meet their social or psychological needs [37]. In some studies, it also refers to the gratification sought [38,39]. With the rapid development of social media and its vital role in an individual's daily life, the motivations for using it and its influence have become hot topics for scholars [e.g., 40,41]. However, the research on international students' social media use motivations in the cross-cultural environment is still relatively limited.

The primary motivations for international students to use social media include information sharing, social interaction, and entertainment [12,17]. Information seeking involves using social media to obtain necessary and relevant information such as daily life updates, academic resources, health, finance, and housing [13,42]. Social motivations encompass using social media to connect with family, friends, and new acquaintances in the host country, facilitating social support and interaction [43]. Entertainment motivations involve using social media to relax, pass the time, and relieve the stress [44].

## Social media use motivations and psychological adaptation

The influence on an individual's well-being is determined by the function of social media rather than the amount of time spent [45]. However, the existing research on whether different types of social media use are related to psychological outcomes is insufficient [46]. A study on Chinese students in Singapore suggested that online information, including daily life information and academic resources, facilitated their adaptation [47]. Based on in-depth interview data from 17 Asian students in the United States, Xie and Chao (2022) found that information obtained from social media can provide social support for international students [13]. Accessing informational support through social media enhances life satisfaction and fosters well-being [48]. Additionally, social media can help international students participate in the host society because it allows international students to be informed about local activities [49], thereby reducing loneliness. Similarly, Guo et al. (2014) found that Chinese international students in Japan utilized social media for social-informational purposes, which enhanced their life satisfaction and contributed to a sense of happiness [48].

"Social" is the essence of social media [50]. For international students, social media provides a way to connect with families and friends in their home countries. Family and friends can provide emotional and informational support to reduce international students' stress [14,51,52]. Peng (2016) also indicated that Chinese mainland students in Hong Kong connect with family and friends through social media mainly to meet their emotional needs, thus promoting adaptation and life experience [53]. International students posting writings or photos on social media about their new life in the host country can strengthen connections with people back home, thereby facilitating their adaptation process [54].

Furthermore, social media also provides a channel for international students to interact with locals, which avoids social isolation [7]. A longitudinal study of international students in 32 countries confirmed that using Facebook to interact with host countries reduced homesickness [55]. Similarly, Hofhuis et al. (2019) suggested that international students who used social media to strengthen their ties with locals could gain social support and participate in the host country, thereby increasing life satisfaction [56]. Moreover, video chat via social media can help international students find like-minded people and reduce stress in a new culture [57].

In contrast, the relationship between entertainment motivations and psychological adaptation is unclear. Social media provides international students with various entertainment options to pass their time [54], alleviate stress and relaxation [13], and enhance life satisfaction [58]. Engaging in activities such as playing games and other forms of entertainment on social media can enable individuals to establish and develop social relationships, leading to the acquisition of social capital [59]. The positive effects of social capital on psychological well-being have been demonstrated [16,60]. However, using social media to obtain entertainment content may lead to problematic social media use [61]. Besides, the entertainment function provided by social media distracts international students' attention [62] and increases loneliness [48].

To summarize, scholars are consistent in the view that information seeking motivations and social motivations play a positive role in psychological adaptation, but whether entertainment use can promote psychological adaptation remains to

be seen. This study argues that entertainment motivations are beneficial in alleviating the loneliness and anxiety of international students in an unfamiliar environment. Thus, the following hypotheses are suggested:

*H1: There is a positive relationship between information seeking motivations and Chinese students'psychological adaptation in Malaysia.*

*H2: There is a positive relationship between social motivations and Chinese students' psychological adaptation in Malaysia.*

*H3: There is a positive relationship between entertainment motivations and Chinese students' psychological adaptation in Malaysia.*

## Acculturation orientations and psychological adaptation

An individual's acculturation orientations often influence his or her beliefs, attitudes, and behaviors [63]. There has been a scarcity of research examining the relationship between mainstream orientations and psychological adaptation [32], and the existing studies have produced inconsistent findings. A meta-analysis of 325 studies argued that mainstream orientations were a double-edged sword for psychological adaptation, which can have positive and negative effects [64]. Individuals who hold mainstream orientations are more likely to appreciate the lifestyle of the host society, follow local customs, and have more social interactions with local people, thus improving their psychological adaptation [65]. Besides, mainstream orientations may reduce discrimination and stereotypes of international students and thus experience less conflict of values [66]. Chinese students' attitudes towards Islamic culture positively influenced their psychological adaptation in Malaysia [67]. An online survey involving Chinese immigrants in New Zealand revealed that their orientations towards New Zealand culture could serve as a predictor of their happiness levels [22]. Moreover, mainstream orientations predict less adaptation stress [34,68], improved self-esteem and life satisfaction [33], and less depression [69].

Conversely, some studies have failed to reveal any significant relationship between host cultural orientation and psychological adaptation [9,70]. Moreover, most international students may return to their home country after graduation, so they consciously maintain their ethnic orientations [71]. Thus, mainstream orientations may lead individuals to coordinate the two cultures and enhance their stress of dealing with two cultural norms [72]. Different values and norms may lead to information pressure and psychological conflict [73] and reduce well-being [74].

Therefore, to study the correlation between acculturation orientations and psychological adaptation, it is necessary to consider the cultural differences between the host and home countries [66]. This study is aimed at Chinese students studying in Malaysia. As Asian countries, Malaysia and China are physically and culturally close to each other [75]. In addition, the Chinese community in Malaysia has retained many cultural practices that are similar to Chinese traditions, including language, cuisine and festivals [76]. Thus, Chinese students may feel less pressure when faced with two cultures. This study assumes that Malaysian cultural orientations are positively related to psychological adaptation.

*H4a: There is a positive relationship between Malaysian cultural orientations and Chinese students' psychological adaptation in Malaysia.*

Similarly, previous studies have yielded varied and inconclusive findings regarding the correlation between ethnic orientations and psychological adaptation. On the one hand, ethnic cultural orientations have been proven helpful in psychological adaptation [32]. Individuals with strong ethnic orientations often experience a heightened sense of group belonging, which contributes to their mental health [77] and self-esteem [12]. Likewise, Du and Lin (2019) stated that Chinese individuals with high Chinese cultural orientations were happier in New Zealand than those with a low Chinese cultural orientation [22].

However, Chinese cultural orientations of Chinese students in the United States have no significant impact on their mental health [78]. More recently, Zheng and Ishii (2023) revealed that Chinese students' Chinese cultural orientations negatively influenced their psychological adaptation in the United States, primarily attributed to their strong reliance on emotional support from their home country. While such distant emotional support may alleviate loneliness, it weakens their connection to the host culture and increases stress [9]. This phenomenon was attributed to a high level of ethnic orientations that can contribute to a higher stress level when individuals encounter two distinct cultural environments [69].

Therefore, the relationship between ethnic orientations and psychological adaptation is also influenced by the cultural distance between the host and the home country. When the differences between the two cultures are small, there may be less pressure for international students to maintain their ethnic orientations during the adaptation process. Given the proximity and cultural similarities between Malaysia and China, the following hypothesis is derived:

*H4b: There is a positive relationship between Chinese cultural orientations and Chinese students' psychological adaptation in Malaysia.*

## Social media use motivations and acculturation orientations

Social media emerged as a convenient and expansive platform, enabling individuals to broaden their exposure to the host culture, thereby fostering a deeper appreciation and understanding of its values, norms, practices, and associated knowledge [79]. Individual cultural values are maintained and strengthened through continuous online information access and social interaction [34]. Social media could promote cultural value systems through information exchange [80]. Drawing upon in-depth interviews with 12 highly educated Chinese individuals in Western countries, Mao and Qian (2015) suggested that Facebook provided access to mass media information and facilitated a deeper understanding of the host culture. Furthermore, the study highlighted that information shared by Facebook contacts served as a crucial source of information about social and cultural events in the host country, enabling Chinese individuals to be exposed to new cultures [81].

Moreover, social media provides a convenient channel for accessing host country cultures that are difficult to reach in daily life [82]. For instance, viewing documentaries about host countries on social media can help individuals better understand local viewpoints, minimizing the risk of offensive actions, and cultural misinterpretations [82]. Rich resources on social media present Chinese culture from multiple perspectives, which can help Southeast Asian students better learn Chinese language, understand cultural differences, and reduce cultural conflicts [83]. Thus, gaining cultural insights through social media can effectively reshape and mitigate existing stereotypes international students have about their host countries [84]. The individual using social media may contribute to fostering greater acceptance of people from diverse backgrounds [85].

On social media, everyone can be a publisher of information, which contributing to a greater understanding of diverse viewpoints and perspectives [84]. For international students, social media has become a bridge for them to establish friendship and exchange information with local people, which greatly enhances their understanding and identification of the host culture [86]. Therefore, utilizing social media can potentially facilitate greater acceptance towards individuals from various cultural backgrounds [85]. Additionally, social media serves as a powerful tool for foreign language learning enables international students to learn new words, correct grammatical mistakes, and improve their language proficiency by communicating with locals [49,85]. To sum up, social media provides an excellent platform for international students to understand and explore different cultures, enabling them to continuously broaden their cultural horizons and deepen their awareness and appreciation of diverse cultures [87].

The above findings highlight using social media to seek information and social interaction positively influences the mainstream orientations. However, there is a dearth of research exploring the link between entertainment motivations and mainstream orientations. Watching movies about Australia can help Asian international students understand Australian

culture and better integrate into Australia [88]. Likewise, subtitles, shows, films, and other entertaining videos on YouTube provide insights into cultural norms and practices that help shape the mainstream orientation of international students [7]. Drawing from the aforementioned studies, the following hypotheses were proposed:

*H5a: There is a positive relationship between information seeking motivations and Malaysian cultural orientations.*

*H6a: There is a positive relationship between social motivations and Malaysian cultural orientations.*

*H7a: There is a positive relationship between entertainment motivations and Malaysian cultural orientations.*

However, limited research exists regarding the connection between motivations for social media use and ethnic orientations. International students can use social media to maintain ties with their home culture while living, studying, and socializing abroad [89]. For example, social media allows international students to stay informed about the latest discussions in the social, cultural, and political spheres of their home countries [7]. Drawing upon data gathered from 12 focus groups involving 49 immigrants, scholars proposed that exposure to information from participants' countries of origin was linked to their aspiration to maintain their ethnic orientations [90].

In addition, interacting online and sharing information with people from one's home country could help strengthen ethnic cultural beliefs and values [34]. Social media enables international students to stay in touch with their families and friends at home and to access the most recent updates about their home country, thus reinforcing their ethnic orientations [43]. Likewise, video chat allows international students to engage in shared activities and spend time together, thereby fostering a stronger sense of connection [57]. Thus, social media help international students feel like they are still part of their home country [91]. Social contact with family and friends were essential predictors of ethnic orientations [92].

However, there is a scarcity of research examining the relationship between entertainment motivations and ethnic orientations. International students can access entertainment content produced by their home country through digital technology [13,88]. Entertainment content fosters a feeling of connectedness to one's ethnic culture by showcasing unique themes, languages, and cultural traits of ethnic cultures [13]. Taken together, this study proposes the following hypotheses:

*H5b: There is a positive relationship between information seeking motivations and Chinese cultural orientations.*

*H6b: There is a positive relationship between social motivations and Chinese cultural orientations.*

*H7b: There is a positive relationship between entertainment motivations and Chinese cultural orientations.*

## The mediating role of acculturation orientations

Acculturation orientations are shaped by various factors, including social interactions, media consumption, and personal experiences, which collectively influence adaptation outcomes [66]. As international students engage with social media, they are exposed to cultural content and interactions that can enhance their connections with the host culture. This strengthened mainstream orientations, in turn, facilitates psychological adaptation by providing a sense of belonging and reducing cultural stress. In other words, acculturation orientations can mediate the relationship between the predictive factors and outcomes.

An earlier study of 280 international students in the United States showed that computer-mediated communication impacted psychological adaptation through acculturation orientations [93]. Du and Lin (2019) highlighted the mediating role of New Zealand cultural orientations in the relationship between social media usage and the psychological well-being of Chinese individuals residing in New Zealand [22]. Drawing upon the previous discussion, this study suggests that social

media use motivations would improve Malaysian and Chinese cultural orientations and thus benefit the psychological adaptation of Chinese students in Malaysia. Therefore, the following hypotheses are proposed:

*H8a: The relationship between information seeking motivations and Chinese students' psychological adaptation is mediated by Malaysian cultural orientations.*

*H8b: The relationship between information seeking motivations and Chinese students' psychological adaptation is mediated by Chinese cultural orientations.*

*H9a: The relationship between social motivations and Chinese students' psychological adaptation is mediated by Malaysian cultural orientations.*

*H9b: The relationship between social motivations and Chinese students' psychological adaptation is mediated by Chinese cultural orientations.*

*H10a: The relationship between entertainment motivations and Chinese students' psychological adaptation is mediated by Malaysian cultural orientations.*

*H10b: The relationship between entertainment motivations and Chinese students' psychological adaptation is mediated by Chinese cultural orientations.*

To sum up, the conceptual framework constructed in this study is shown in Fig 1.

## Method

### Respondents

The study was approved by the Ethics Committee for Research Involving Human Subjects from the Universiti Putra Malaysia. Informed consent was obtained from the respondents before they filled out the questionnaire to ensure that all

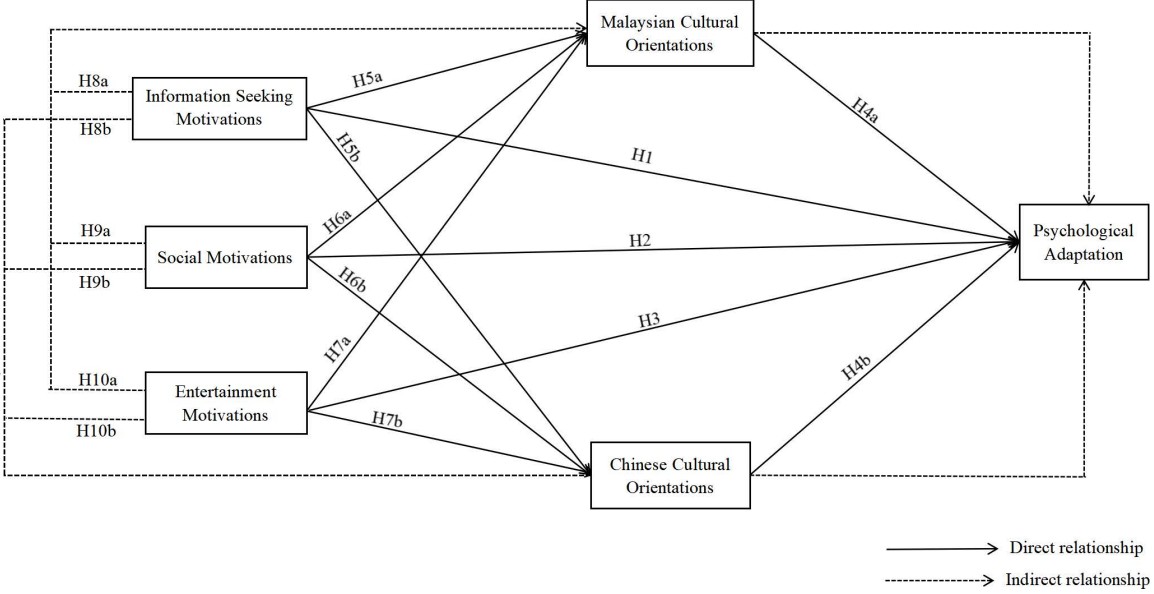

**Fig 1. A proposed conceptual framework for psychological adaptation.**

respondents were willing to participate in the research voluntarily. Respondents were only permitted to proceed with the questionnaire after they had reviewed and explicitly agreed to the content of the informed consent form by clicking the "agree" button. Additional information regarding the ethical, cultural, and scientific considerations specific to inclusivity in global research is included in the Supporting Information (SX Checklist).

Due to Chinese students' broad distribution across Malaysian institutions and large geographical span, generating a comprehensive sampling frame is difficult. Additionally, the specific number of Chinese students at each university is unknown to the researcher. Thus, probability sampling is unsuitable for this study. Instead, the study adopted the convenience sampling method.

With the proliferation of the internet and social media, online surveys have become an efficient and convenient method of data collection. Compared with the traditional method (e.g., paper-pencil), online questionnaire distribution is a very time-saving method. It can reach a broader range of samples, thereby facilitating the collection of a larger sample size [94]. Specifically, the respondents were recruited through Wenxing, an online questionnaire survey platform widely used in China. The survey took place from November to December 2022, and links to the questionnaire were distributed on social media platforms among Chinese international students pursuing undergraduate or postgraduate degrees at higher education institutions in Malaysia. In this way, respondents were guaranteed to be social media users. According to scholars, respondents had to have lived in Malaysia for at least six months when studying cross-cultural research [95]. Based on Taro Yamane's formula (2003), where $n = N/(1 + N(e)^2)$ [96], with a population (N) of approximately 20,000 and a sample error (e) of 5%, the minimum required sample size is calculated to be 395. Generally, sample sizes between 30 and 500 are considered appropriate for most studies [97]. Ultimately, 438 students participated in the study.

Of the 438 respondents, 55.3% were women and 44.7% were men. Most of the respondents (75.8%) were unmarried. Additionally, 48.2% of those surveyed were enrolled in bachelor's degree programs. The average age of the participants was 25.70 years (SD = 5.81), and they had been residing in Malaysia for an average of 14.58 months (SD = 9.75).

## Measures

The questionnaire consists of two parts. The first part is demographic information, including age, gender, education level, English proficiency, and the duration of their stay in Malaysia. The second part of the questionnaire is the measurement of the main variables, using a five-point Likert scale ranging from "strongly disagree (1)" to "strongly agree (5)." The scales were adapted from previous research with minor adjustments to ensure they are well-suited to the specific context of this study. Table 1 provides a comprehensive list of all measurement items along with their respective literature sources.

## Statistical analysis

In this study, the data analysis was carried out using SPSS 26 and AMOS 26, encompassing both descriptive and inferential analysis. Descriptive analysis specifically involved calculating the mean and standard deviation, as well as correlation analysis among the variables. Confirmatory factor analysis (CFA) was used to investigate the validity and reliability of the questionnaire. Path analysis identified the relationship between social media use motivations, acculturation orientations, and psychological adaptation.

## Findings

**Preliminary analysis.** When performing Structural Equation Modeling (SEM) analysis, it is essential to examine the skewness and kurtosis of the variables to verify compliance with the normal distribution assumption [106]. If the absolute values of skewness and kurtosis fall within the range of 0–1, the data can be considered approximately normally distributed [107]. In the current study, the absolute values of skewness and kurtosis are below 1 (see Table 2), indicating that the data in the current study were normally distributed. Additionally, evaluating multicollinearity is essential before

**Table 1. Measurement items.**

| Latent Construct | Items | References |
|---|---|---|
| Information Seeking Motivations | In Malaysia, I use social media to find information. | [98,99] |
| | In Malaysia, I use social media to get information for free. | |
| | In Malaysia, I use social media to do research. | |
| Social Motivations | In Malaysia, I use social media to keep in touch with friends in China. | [100,101] |
| | In Malaysia, I use social media to keep in touch with local people. | |
| | In Malaysia, I use social media to meet new friends. | |
| | In Malaysia, I use social media to keep in touch with families in China. | |
| Entertainment Motivations | In Malaysia, I use social media as a source of entertainment. | [100,102] |
| | In Malaysia, I use social media for fun. | |
| | In Malaysia, I use social media to relax. | |
| | In Malaysia, I use social media to pass the time. | |
| Malaysian cultural orientations | I enjoy engaging in cultural activities with Malaysian people. | [103,104] |
| | I enjoy Malaysian cultural entertainment (e.g., movies, music) | |
| | It is important for me to maintain or develop Malaysian cultural practices. | |
| | I believe in Malaysian values. | |
| | I am interested in having Malaysian friends. | |
| | I enjoy speaking Malay. | |
| | I like to celebrate holidays in the Malaysian way. | |
| | I am comfortable socializing with a group of Malaysians who do not speak Chinese. | |
| Chinese Cultural Orientations | I enjoy engaging in cultural activities with people from China. | [103,104] |
| | I enjoy Chinese cultural entertainment (e.g., movies, music). | |
| | It is important for me to maintain or develop Chinese cultural practices. | |
| | I believe in Chinese values. | |
| | I am interested in having Chinese friends. | |
| | I enjoy speaking Chinese. | |
| | I like to celebrate holidays in the Chinese way. | |
| | I am comfortable socializing with a group of people from China. | |
| Psychological Adaptation | I feel excited about being in Malaysia. | [105] |
| | I feel out of place like I don't fit into Malaysian culture. | |
| | I feel sad to be away from my family and friends in China. | |
| | I feel sad to be away from my family and friends in China. | |
| | I feel lonely without my family and friends around me in Malaysia. | |
| | I feel homesick when I think of China in Malaysia. | |
| | I feel frustrated by the difficulties encountered in Malaysia. | |
| | I feel happy with my day-to-day life in Malaysia. | |

**Table 2. Descriptive statistics of the latent constructs (N=438).**

| Constructs | Mean | SD | Skewness | Kurtosis | VIF | TOL |
|---|---|---|---|---|---|---|
| Information Seeking Motivations | 2.98 | .77 | 0.10 | -0.46 | 1.22 | 0.82 |
| Social Motivations | 3.20 | .73 | -0.23 | -0.42 | 1.24 | 0.81 |
| Entertainment Motivations | 3.15 | .78 | -0.12 | -0.52 | 1.07 | 0.93 |
| Malaysian Cultural Orientations | 3.28 | .71 | -0.11 | -0.48 | 1.33 | 0.75 |
| Chinese Cultural Orientations | 3.80 | .76 | -0.67 | 0.10 | 1.04 | 0.96 |

using SEM to analyze data. The common method to identify multicollinearity issues is to check the variance inflation factor (VIF) and tolerance (TOL) values. A variable with a VIF value greater than ten and a TOL value less than 0.10 indicates that the variable is highly correlated with other variables [106]. As the data shown in Table 1, all the VIF values were less than 10, and the TOL value was higher than 0.1. Thus, multicollinearity was not an issue for this study.

**Measurement model analysis.** The results of CFA are displayed in Fig 2. According to the threshold value suggested by Byrne (2016) [108], the results suggest a good model fit with the data. Specifically, $X^2/(df)=1.258$, which is smaller than

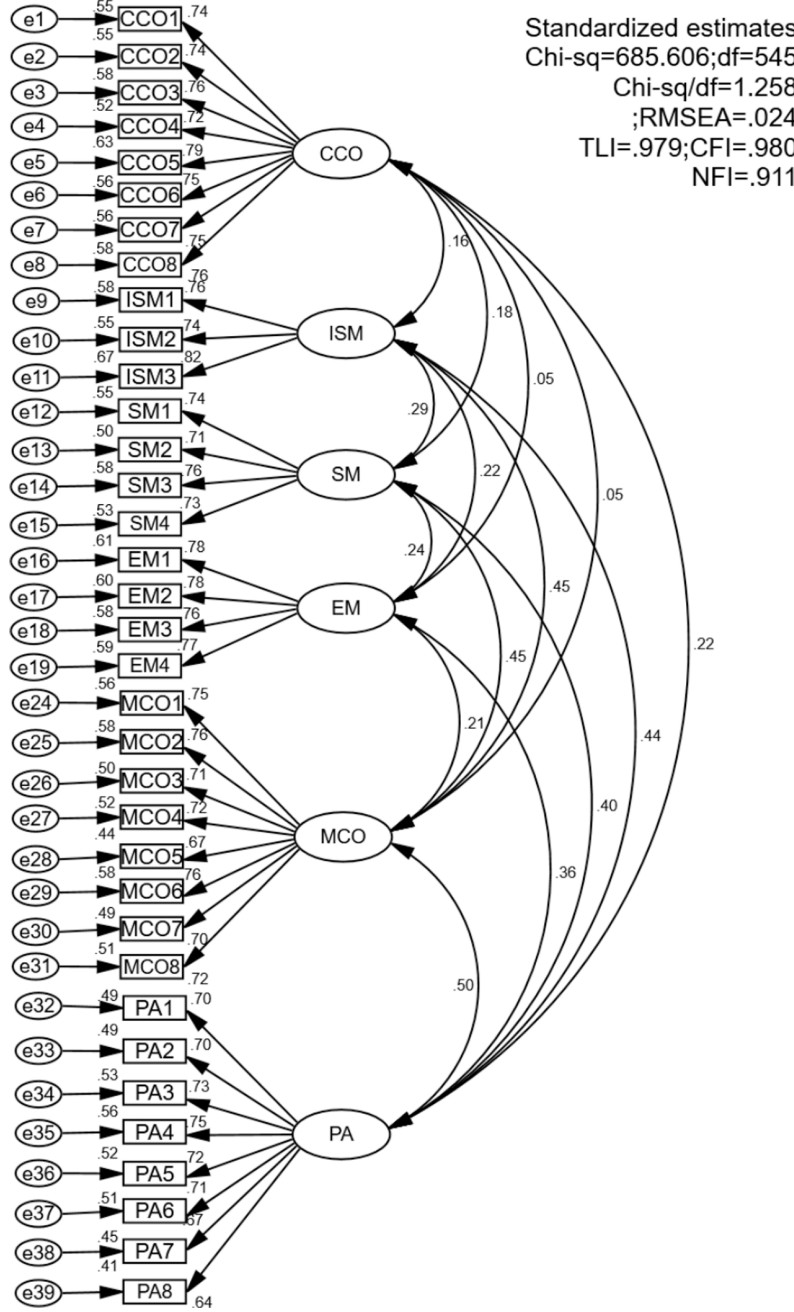

**Fig 2. Measurement model with standardized estimates.**

the accepted value of 3; CFI = .980 and TLI = .979, which are larger than the accepted value of .90; RMSEA = .024, which is smaller than the cut-off points of .08.

As indicated in Table 3, both the composite reliability (CR) and Cronbach's alpha for each measurement exceeded the threshold of 0.7. Most of the average variance extracted (AVE) of the variables was greater than 0.5 and the square root of AVE was greater than the correlation coefficient among the variables. The above results indicate that the reliability, convergent validity, and discriminant validity of the construct are not a concern.

It is critical to check common method bias while collecting data through self-reported surveys or questionnaires [109], especially data on dependent and independent variables from the same respondents [110]. Harman's single-factor test is controversial but widely used to identify common method bias [111]. In this study, a total of 35 items representing the eight variables were included in an exploratory factor analysis. Only 24.54% of the variance was accounted for by a single factor, indicating that common method bias was not a significant concern.

**Assessment of the structural model.** Fig 3 shows that the $R^2$ value on psychological adaptation is 0.396, which suggests that information seeking motivations, social motivations, entertainment motivations, Malaysian cultural orientations, and Chinese cultural orientations contributed to around 40% of the variance in psychological adaptation. According to the criteria of the $R^2$ value proposed by Cohen (1988), the result represents a substantial predictive accuracy [112].

## Hypotheses testing

According to Table 4, information seeking motivations ($\beta$ = .195, $p$ < .001), social motivations ($\beta$ = .131, $p$ = .019), and entertainment motivations ($\beta$ = .215, $p$ < .001). Malaysian cultural orientations ($\beta$ = .299, $p$ < .001), and Chinese cultural orientations ($\beta$ = .135, $p$ = .003) all significantly contributed to psychological adaptation. Thus, H1, H2, H3, H4a, and H4b were supported. Information seeking motivations ($\beta$ = .340, $p$ < .001) and social motivations ($\beta$ = .341, $p$ < .001) have positive effects on Malaysian cultural orientations, H5a and H6a were supported. However, the influence of entertainment motivations on Malaysian cultural orientations is not statistically significant ($\beta$ = .051, $p$ = .315), thus rejecting H7a. As for Chinese cultural orientations, information seeking motivations and social motivations significantly showed significant influence ($\beta$ = .118, $p$ = .044; $\beta$ = .145, $p$ = .015 respectively). Entertainment motivations had no influence ($\beta$ = -.016, $p$ = .782). Thus, H5b and H6b were supported, and H7b was rejected.

A total of 5000 bootstrap samples were used to evaluate the mediating effect. When the 95% confidence interval (CI) does not include zero, it indicates a significant indirect effect [106]. As shown in Table 6, when controlling Malaysian cultural orientations, the indirect effect of information seeking motivations on psychological adaptation was significant ($b$ = .094, 95% CI [.057, .150], $p$ < .001), which suggests that Malaysian cultural orientations mediated the relationship

**Table 3. Reliability and validity of the measurement (N = 438).**

| Latent Construct | Convergent validity | | Discriminant validity | | | | | | Reliability | |
|---|---|---|---|---|---|---|---|---|---|---|
| | Loadings | AVE | ISM | SM | EM | MCO | CCO | PA | CR | Cronbach's alpha |
| ISM | 0.74-0.82 | 0.60 | **.77** | | | | | | 0.82 | 0.82 |
| SM | 0.71-0.76 | 0.54 | .24** | **.74** | | | | | 0.83 | 0.82 |
| EM | 0.76-0.78 | 0.59 | .19** | .20** | **.77** | | | | 0.85 | 0.85 |
| MCO | 0.67-0.76 | 0.52 | .39** | .39** | .18** | **.72** | | | 0.90 | 0.90 |
| CCO | 0.72-0.79 | 0.57 | .14** | .16** | .04 | .05 | **.76** | | 0.91 | 0.91 |
| PA | 0.64-0.75 | 0.50 | .37** | .34** | .31** | .45** | .20** | **.71** | 0.89 | 0.89 |

Note: (1) ISM = Information Seeking Motivations; SM = Social Motivations; EM = Entertainment Motivations; MCO = Malaysian cultural orientations; CCO = Chinese cultural orientations; PA = Psychological Adaptation; (2) **$p$ < .01.

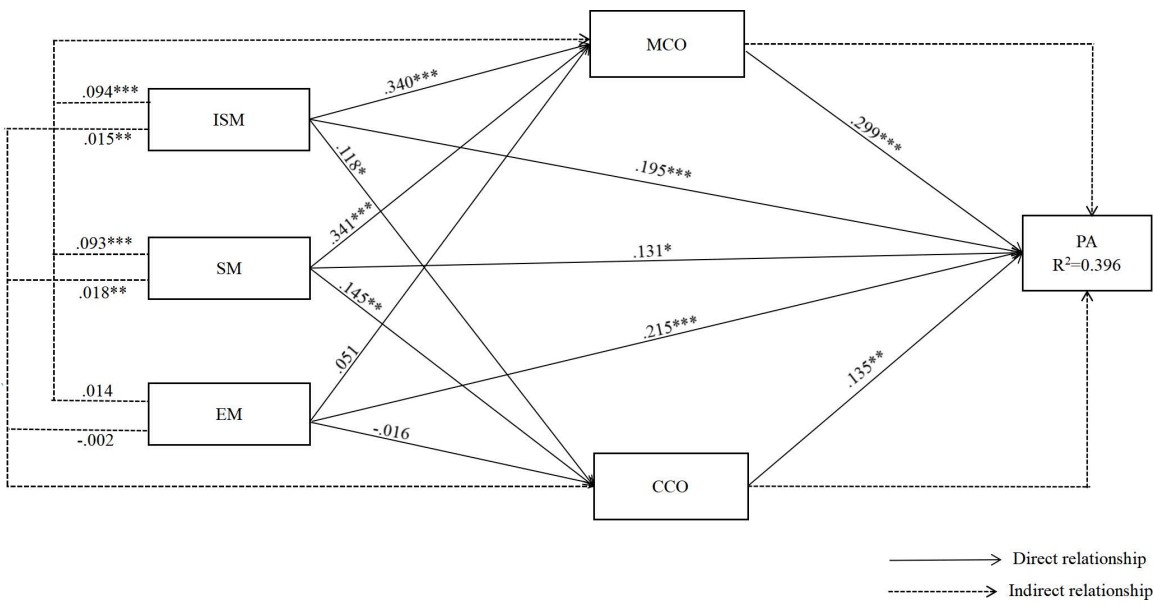

**Fig 3. Structural model with standardized estimate.**

**Table 4. Results of hypotheses testing.**

| Hypothesis (direct paths) | | Standardized coefficient | S.E. | C.R. | p-value | Results |
|---|---|---|---|---|---|---|
| H1 | ISM→PA | .195 | .052 | 3.497 | *** | Supported |
| H2 | SM→PA | . 131 | .050 | 2.352 | .019 | Supported |
| H3 | EM→PA | .215 | .045 | 4.332 | *** | Supported |
| H4a | MCO→PA | .299 | .050 | 5.036 | *** | Supported |
| H4b | CCO→PA | .135 | .039 | 2.958 | .003 | Supported |
| H5a | ISM→MCO | .340 | .061 | 6.074 | *** | Supported |
| H5b | ISM→CCO | .118 | .064 | 2.017 | .044 | Supported |
| H6a | SM→MCO | .341 | .061 | 6.019 | *** | Supported |
| H6b | SM→CCO | .145 | .063 | 2.439 | .015 | Supported |
| H7a | EM→MCO | .051 | .054 | 1.005 | .315 | Rejected |
| H7b | EM→CCO | -.016 | .060 | -0.276 | .782 | Rejected |
| **Hypothesis (indirect paths)** | | **Standardized coefficient** | **95% CI** | | **p-Value** | **Results** |
| | | | Lower | upper | | |
| H8a | ISM→MCO→PA | .094 | .057 | .150 | *** | Supported |
| H8b | ISM→CCO→PA | .015 | .001 | .038 | .033 | Supported |
| H9a | SM→MCO→PA | .093 | .051 | .153 | *** | Supported |
| H9b | SM→CCO→PA | .018 | .003 | .044 | .014 | Supported |
| H10a | EM→MCO→PA | .014 | -.010 | .045 | .249 | Rejected |
| H10b | EM→CCO→PA | -.002 | -.019 | .013 | .759 | Rejected |

Note: (1) ISM = Information Seeking Motivations; SM = Social Motivations; EM = Entertainment Motivations; MCO = Malaysian cultural orientations; CCO = Chinese cultural orientations; PA = Psychological Adaptation; (2) ***p < .01.

between information seeking motivations and psychological adaptation. Similarly, the indirect effect of social motivations on psychological adaptation was significant (b = .093, 95% CI [.051,.153], p < .001). Differently, the indirect effect of entertainment motivations on psychological adaptation was not significant (b = .014, 95% CI [-.010,.045], p = .249), which suggests that Malaysian cultural orientations did not mediate the relationship between entertainment motivations and psychological adaptation. Thus, H8a and H9a were supported, and H10a was rejected.

When controlling Chinese cultural orientations, the indirect effect of information seeking motivations on psychological adaptation was significant (b = .015, 95% CI [.001,.038], p = .033), which suggests that Chinese cultural orientations mediated the relationship between information seeking motivations and psychological adaptation. Similarly, Chinese cultural orientations served as a mediator in the relationship between social motivations and psychological adaptation (b = .018, 95%CI [.003,.044], p = .014). On the contrary, the indirect effect of entertainment motivations did not exist (b = -.002, 95%CI [-.019,.013], p = .759). Thus, H8b and H9b were supported, while H10b were rejected.

## Discussion

This study explores the influence of social media use motivations on Chinese international students' psychological adaptation in Malaysia through Malaysian cultural orientations and Chinese cultural orientations.

H1, H2, and H3 demonstrate the beneficial effect of different social media use motivations on enhancing psychological adaptation. Consistent with Liu et al. (2018), information obtained on social media, especially information about the host country is benefit to improving life satisfaction and thus enhancing psychological adaptation [79]. Connecting with family and friends in the home country or communicating with host country on social media benefits psychological adaptation. For example, communicating with people in the host countries can reduce intercultural barriers and strengthen a sense of belonging [113]. Consistent with Sin and Kim (2013), using social media can meet the information needs of daily life and is conducive to cross-cultural adaptation [114]. Existing studies have inconsistent conclusions about the effects of entertainment motivations on psychological adaptation, this study suggested that entertainment usage also contributes to psychological adaptation. The result is consistent with previous research [17,59], indicating that entertainment activities on social media can relieve international students from pressures in the cross-cultural process.

H4a suggests that Malaysian cultural orientations significantly influence psychological adaptation. The result aligns with Schotte et al. (2018), who indicated a positive relationship between host cultural orientations and psychological adaptation [32]. The possible explanation is that understanding the host country's cultural rules can experience less stress [68] and improve life satisfaction [65]. Similarly, H4b indicates that a high degree of Chinese cultural orientations are beneficial for the psychological adaptation of Chinese students in Malaysia. The main reason is that ethnic cultural orientations can enhance positive emotions such as self-esteem [12], happiness [22], and well-being [77]. Compared with Chinese cultural orientations, Malaysian cultural orientations are more conducive to improving psychological adaptation. This is consistent with the conclusion of Du & Lin (2019) that host cultural orientations are more conducive to improving Chinese immigrant's happiness [22].

Information seeking motivations and social motivations have a significant influence on Malaysian and Chinese cultural orientation, while entertainment motivations have no effect on either. Therefore, H5a, H5b, H6a, and H6b are supported, and H7a and H7b are rejected. Information usage helps to increase mainstream and ethnic orientations, which is consistent with Qiu et al. (2013) [80]. Continuous social interaction can enhance people's understanding of different cultures and influence cultural orientation [34,84]. As Liu et al. (2018) indicated that rich information and powerful social functions of social media allow people to easily and quickly understand the values and norms of the host culture [79]. Additionally, international students can maintain contact with their home culture, engaging in social interactions, discussions, and exchanges with people of the motherland, which reinforces their sense of belonging and identification with their ethnic culture [115].

Notably, entertainment usage had no significant effect on two types of cultural orientations, potentially attributed to several factors. Firstly, Asian students prefer American, Korean, and Japanese entertainment content [88]. Thus, it is plausible that entertainment motivations do not significantly influence Malaysian and Chinese cultural orientations. Secondly, social media platforms often tend to prioritize popular or trending content under the influence of algorithmic recommendations [116,117], which may not always adequately reflect cultural diversity and uniqueness. This distribution mechanism may lead to a narrow range of entertainment content that users are exposed to on social media, thus limiting their understanding and experience of different cultures. Furthermore, digital media has accelerated the exchange and integration of global cultures, fostering a trend of cultural homogenization worldwide [118]. Cultural homogeneity may lead to the disappearance of local cultural characteristics [119].

H8, H9, and H10 demonstrate that acculturation orientations can explain how social media use motivations affect psychological adaptation. This also suggests that the effect of social media use motivations on psychological adaptation is not direct but happens through some intermediate mechanism. Specifically, in the influence of information and social motivations on psychological adaptation, both Malaysian and Chinese cultural orientations play a mediating role. However, the two cultural orientations did not mediate the relationship between entertainment motivations and psychological adaptation. Therefore, the extent to which Chinese international students use social media for entertainment motivations is directly associated with their psychological adaptation in Malaysia without a significant indirect effect through Malaysian and Chinese cultural orientations. This is attributable to the fact that entertainment usage had no effect on cultural orientations. Therefore, cultural orientations cannot serve as an effective mediator to explain the relationship between entertainment motivations and psychological adaptation.

## Limitations and implications

The study had several major limitations that had to be addressed. First, the study only focused on the Malaysian context, which makes its findings difficult to apply to Chinese students studying in other Asian countries. Again, as the study only included Chinese students, it is not representative of international students from other countries in Malaysia. To achieve more generalized results, it is crucial to examine more diverse and representative samples. Second, the research methods used in this study are insufficient. The study used a cross-sectional design and surveyed participants only at one point in time. Cross-sectional design is not effective in establishing causal relationships between variables [120]. Similarly, since the experience of overseas students is a dynamic process, longitudinal analysis can draw more effective conclusions. Third, although quantitative studies can identify the relationship between variables, everyone's social media use behaviors and cross-cultural experiences differ. Qualitative research explores diverse experiences and narratives from an individual perspective, offering comprehensive, detailed, and dynamic descriptions, thereby revealing the underlying meanings of complex phenomena [121].

This study has three main implications. First, this research makes substantial contributions to the field of cross-cultural adaptation. It stands as one of the pioneering studies to explore the relationship between social media use motivations, acculturation orientations, and the psychological adaptation of Chinese international students within Asian countries. By emphasizing the significance of considering mediators when assessing the influence of social media usage on psychological adaptation, this study facilitates a deeper understanding of the underlying mechanisms that influence the cross-cultural adaptation process of international students. Furthermore, by drawing the attention on the Asian context—an area that has received relatively scant attention compared to Western contexts—this research enriches the geographical diversity of cross-cultural adaptation literature. It establishes a more comprehensive and diversified landscape in the domain of cross-cultural adaptation studies.

Second, this study also generates implications for developing acculturation theory. Acculturation theory posits that host and ethnic orientations are independent dimensions, and an individual's attitude towards the two cultures influences adaptation outcomes. This study verifies the positive relationship between the two cultural orientations and psychological

adaptation, which supports the applicability of this theory in international students' groups. Besides, this study applies the theory to a new context and enriching the research results of the theory. Furthermore, this study demonstrates that social media use motivations influence cultural orientations, which helps to expand the current research in the social media area. As media technology rapidly advances and facilitates increased contact with different cultures, there is a growing need to integrate media use into the study of acculturation theory.

Third, in terms of practical implications, the study reminds Chinese students to identify their goals and needs and engage in social media use behaviors purposefully, such as getting information to solve various problems, building their social networks with different people, learning Malaysian cultural knowledge, and participating in entertainment activities to relieve stress in the cross-cultural process. Besides, the study provides valuable insights for social media designers and managers. The results show that using social media for information seeking, social, and entertainment motivations yields positive influences on the psychological adaptation of international students. Therefore, if the designers of social media can develop novel applications to enhance these functions and services, the use of social media may be more helpful to international students in the adaptation process. To be specific, designers should develop features that streamline the process of finding relevant and accurate information for international students. This could involve integrating advanced search functionalities, creating dedicated content hubs for international students, or partnering with educational institutions and local organizations to provide curated content. In view of the fact that entertainment use has a significant effect on psychological adaptation, but has no effect on cultural orientation. Thus, social media designers can incorporate diverse and culturally relevant content, such as cultural variety shows, short videos, interactive games, and quizzes, which can promote cultural understanding and learning.

Malaysian policymakers and educators should consider using popular social platforms such as WeChat and WhatsApp to connect with Chinese students to learn about their needs and offer support. Considering the positive impact of Malaysian cultural orientations on psychological adaptation, relevant departments can set up a special section for international students on social media platforms. This section would serve as a hub for disseminating knowledge about Malaysian culture and values, facilitating exchanges with international students and increasing intercultural understanding.

## Conclusions

This study aims to examine the relationship between motivations for social media use and psychological adaptation considering the mediating role of acculturation orientations. The respondents were Chinese international students studying in Malaysia. Although Chinese international students have long been the focus of cross-cultural scholars, this group has received little attention in the Asian context, especially in Malaysia. The results showed that three types of social media use motivations and two cultural orientations contribute to psychological adaptation. Information seeking and social motivations have a significant impact on both Malaysian and Chinese cultural orientation. Entertainment motivations had no effect on either cultural orientations. Accordingly, the mediating mechanism of acculturation orientations in the relationship between social media use motivations and psychological adaptation is also complex. However, the sample was restricted to Chinese international students in Malaysia, which may limit the generalizability of the findings. The specific cultural and contextual factors influencing Chinese students in Malaysia may not be representative of the experiences of international students from other backgrounds or studying in other countries. Therefore, caution should be exercised in extrapolating the results to broader populations.

## Author contributions

**Conceptualization:** zhao wenwen.

**Investigation:** zhao wenwen.

**Methodology:** zhao wenwen.

**Software:** zhao wenwen.

**Visualization:** zhao wenwen.

**Writing – original draft:** zhao wenwen.

**Writing – review & editing:** zhao wenwen.

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
