## [Decision Letter · Decision Letter 0]

8 Jan 2025

PONE-D-24-52159Exploring the Influence of Social Media Use Motivations on Acculturation Orientations and Psychological Adaptation Among Chinese Students in MalaysiaPLOS ONE

Dear Dr. wenwen,

Thank you for submitting your manuscript to PLOS ONE. After careful consideration, we feel that it has merit but does not fully meet PLOS ONE’s publication criteria as it currently stands. Therefore, we invite you to submit a revised version of the manuscript that addresses the points raised during the review process.

**ACADEMIC EDITOR: **

Please address all the comments and suggestions given by the reviewers.

We look forward to receiving your revised manuscript.

Kind regards,

Kashif Ali, PH.D

Academic Editor

PLOS ONE

(2) Please provide additional details regarding participant consent. In the ethics statement in the Methods and online submission information, please ensure that you have specified (i) whether consent was informed and (ii) what type you obtained (for instance, written or verbal, and if verbal, how it was documented and witnessed). If your study included minors, state whether you obtained consent from parents or guardians. If the need for consent was waived by the ethics committee, please include this information.

https://hrmars.com/papers_submitted/17668/social-media-usage-and-cultural-identity-of-chinese-students-in-malaysia.pdf?

In your revision ensure you cite all your sources (including your own works), and quote or rephrase any duplicated text outside the methods section. Further consideration is dependent on these concerns being addressed.

“This work was supported by Humanities and Social Science Project of Guangdong Ocean University: Study on social media use and acculturation of sojourn (Grant No. 030301142302)”

5. We note that your Data Availability Statement is currently as follows: [All relevant data are within the manuscript and its Supporting Information files.]

Reviewers' comments:

Reviewer's Responses to Questions

**Comments to the Author**

1. Is the manuscript technically sound, and do the data support the conclusions?

Reviewer #1: Yes

Reviewer #2: Yes

Reviewer #3: Partly

2. Has the statistical analysis been performed appropriately and rigorously? 

Reviewer #1: Yes

Reviewer #2: Yes

Reviewer #3: No

3. Have the authors made all data underlying the findings in their manuscript fully available?

Reviewer #1: Yes

Reviewer #2: Yes

Reviewer #3: No

4. Is the manuscript presented in an intelligible fashion and written in standard English?

Reviewer #1: Yes

Reviewer #2: No

Reviewer #3: No

5. Review Comments to the Author

Reviewer #1: -The abstract mentions that motivations (information seeking, social, and entertainment) improve psychological adaptation but does not specify which motivations are most influential. Providing this detail could enhance understanding.

-You need to mention the software you used for the study

-The phrase "help Chinese students improve their psychological adaptation in Malaysia" is repeated in slightly different forms. Condensing these repetitions can improve readability.

The upcoming content is very good.

Reviewer #2: Dear author (s) please incorporate with following suggestions:

The study focuses exclusively on Chinese students in Malaysia, limiting the generalizability of findings to other international student groups or host countries. This limitation should be more explicitly acknowledged in the conclusion.

The literature review is comprehensive but leans heavily on studies conducted in Western contexts. Including more references from Asian or Malaysian studies would improve regional relevance.

While hypotheses H7a and H7b are rejected, the manuscript does not adequately explore why entertainment motivations have no significant effect on acculturation orientations. This needs further elaboration in the discussion.

The manuscript notes that respondents use both Chinese (e.g., WeChat) and Malaysian (e.g., WhatsApp) platforms but does not explore how platform-specific usage influences adaptation. This could add depth to the analysis.

Strengthen the discussion of the rejected hypotheses (H7a, H7b, H10a, H10b) by providing theoretical or empirical justifications.

Highlight the potential implications of specific social media platforms (e.g., cultural proximity in WeChat vs. global networking in WhatsApp).

Consider revising the conclusion to emphasize the broader applicability of findings while acknowledging limitations more explicitly.

The visual presentation of findings, such as path analysis and conceptual frameworks, could be more detailed and visually engaging.

There are occasional grammatical errors and awkward phrasing. For instance, "social media use motivations includes three dimensions" should be corrected to "social media use motivations include three dimensions." A thorough language edit is recommended.

Reviewer #3: Peer Review Report

Manuscript Title:Exploring the Influence of Social Media Use Motivations on Acculturation Orientations and Psychological Adaptation Among Chinese Students in Malaysia

Overall Assessment:This manuscript addresses a highly relevant and timely topic, exploring the relationship between social media use motivations, acculturation orientations, and psychological adaptation of Chinese students in Malaysia. The research is well-conceived and contributes to the growing body of literature on cross-cultural adaptation and social media's role in psychological well-being. However, there are several areas that require clarification and improvement to ensure the manuscript meets the scientific and editorial standards expected by this journal.

Strengths

1. Timeliness and Relevance:The topic aligns with current research trends, addressing the psychological adaptation challenges of international students, a growing population globally.

2. Methodological Rigor:The study employs CB-SEM, an advanced statistical method, to investigate the relationships among variables, providing robust empirical support for the hypotheses.

3. Theoretical Contribution:The manuscript expands acculturation theory by incorporating social media use motivations as predictors and exploring their mediated effects on psychological adaptation.

Major Comments

1. Abstract:The abstract should better emphasize the urgency and significance of the research, particularly its relevance to the growing population of international students.Include details on the sampling method and briefly highlight the implications of the findings.Avoid vague language and strengthen the concluding statements to emphasize the study’s practical applications.

2. Language and Grammar:The manuscript contains grammatical and syntactical errors that hinder readability. A thorough language revision is necessary to improve clarity and academic tone. For example. Original: “Information support can enhance life satisfaction and contribute to a sense of happiness.” Revised: “Accessing informational support through social media enhances life satisfaction and fosters well-being.”

3. Methodology: Sampling Method.The manuscript does not clearly specify the sampling technique (e.g., random, convenience). This information is crucial for evaluating the generalizability of the results.COVID-19 Justification.The claim that the questionnaire was distributed online due to the COVID-19 pandemic is not valid for the timeframe of data collection (November–December 2022). The justification should be revised to focus on logistical or methodological advantages. Questionnaire Adaptation.Clearly explain the rationale for adapting the scales and the specific modifications made to ensure validity in the current context.

4. Findings and Preliminary Analysis.Include descriptive statistics (e.g., skewness, kurtosis) to confirm the normality of the data, as CB-SEM requires normally distributed data. Consider presenting descriptive statistics before the measurement model analysis to align with standard reporting practices.

5. Discussion:Expand on the theoretical implications by linking findings more explicitly to acculturation theory and cross-cultural adaptation literature.Practical implications should offer actionable insights for stakeholders such as educators, social media designers, and policymakers.The discussion should explicitly address the rejected hypotheses (e.g., entertainment motivations’ lack of effect) to provide a balanced and comprehensive interpretation of the findings.

6. Reference Formatting:The manuscript currently follows APA 7, which does not adhere to the referencing style required by PLOS ONE. Reformat all references according to the journal’s guidelines.

7. Data Availability:To comply with PLOS ONE policies, ensure that raw data is publicly available or accessible upon request. Include a data availability statement with a repository link or detailed instructions for access.

Recommendation

Minor Revisions:The manuscript has significant potential but requires substantial improvements in methodology, data reporting, discussion, and formatting to meet the standards of publication in this journal.

Detailed Suggestions for Revision:

1. Revise the abstract to emphasize urgency, significance, and methodological rigor.

2. Conduct a thorough language review to eliminate grammatical errors.

3. Clarify the sampling method and revise the justification for online data collection.

4. Add descriptive statistics and discuss the data distribution.

5. Expand the discussion to address rejected hypotheses and practical implications.

6. Reformat references to align with PLOS ONE guidelines.

7. Provide data.

6. PLOS authors have the option to publish the peer review history of their article (what does this mean? ). If published, this will include your full peer review and any attached files.

**Do you want your identity to be public for this peer review?** For information about this choice, including consent withdrawal, please see our Privacy Policy .

Reviewer #1: **Yes: ** Nornajihah Nadia Hasbullah

Reviewer #2: No

Reviewer #3: **Yes: ** xinxiang gao

---

## [Author Response · Author response to Decision Letter 0]

27 Feb 2025

Sincere thanks should be given to the reviewer for the constructive comments and suggestions. I have tried my best to revise the manuscript as shown in the rebuttal letter.

---

## [Decision Letter · Decision Letter 1]

8 Apr 2025

Exploring the Influence of Social Media Use Motivations on Acculturation Orientations and Psychological Adaptation Among Chinese Students in Malaysia

PONE-D-24-52159R1

Dear Dr. wenwen,

We’re pleased to inform you that your manuscript has been judged scientifically suitable for publication and will be formally accepted for publication once it meets all outstanding technical requirements.

Kind regards,

Kashif Ali, PH.D

Academic Editor

PLOS ONE

Additional Editor Comments (optional):

Reviewers' comments:

Reviewer's Responses to Questions

**Comments to the Author**

1. If the authors have adequately addressed your comments raised in a previous round of review and you feel that this manuscript is now acceptable for publication, you may indicate that here to bypass the “Comments to the Author” section, enter your conflict of interest statement in the “Confidential to Editor” section, and submit your "Accept" recommendation.

Reviewer #3: All comments have been addressed

2. Is the manuscript technically sound, and do the data support the conclusions?

Reviewer #3: Yes

3. Has the statistical analysis been performed appropriately and rigorously? 

Reviewer #3: Yes

4. Have the authors made all data underlying the findings in their manuscript fully available?

Reviewer #3: No

5. Is the manuscript presented in an intelligible fashion and written in standard English?

Reviewer #3: Yes

6. Review Comments to the Author

Reviewer #3: I am currently conducting a peer review of your manuscript and have identified two specific areas requiring attention:

Firstly, the provided DOI link (DOI: 10.6084/m9.figshare.28379111) does not appear to contain the dataset referenced in your manuscript. Upon accessing this link, no associated data or files could be located. Ensuring accurate and accessible data sources is essential for transparency, reproducibility, and credibility of your research findings. I strongly recommend verifying the DOI link provided or uploading the dataset correctly to ensure reviewers and future readers can access the necessary information.

Secondly, I suggest standardizing the formatting of all headings throughout the manuscript. Currently, inconsistencies in heading style, capitalization, and numbering negatively affect readability and professional presentation. Adhering to a clear hierarchical structure—such as consistently using bold or italicized fonts, sentence case or title case capitalization, and sequential numbering (e.g., 1., 1.1, 1.1.1)—would significantly improve manuscript clarity. Consistent heading formatting aligns with publication guidelines and enhances the overall coherence and professionalism of the manuscript.

7. PLOS authors have the option to publish the peer review history of their article (what does this mean? ). If published, this will include your full peer review and any attached files.

**Do you want your identity to be public for this peer review?** For information about this choice, including consent withdrawal, please see our Privacy Policy .

Reviewer #3: **Yes: ** Xinxiang Gao

---

## [Editor Report · Acceptance letter]

PONE-D-24-52159R1

PLOS ONE

Dear Dr. wenwen,

I'm pleased to inform you that your manuscript has been deemed suitable for publication in PLOS ONE. Congratulations! Your manuscript is now being handed over to our production team.

Kind regards,

on behalf of

Dr. Kashif Ali

Academic Editor

PLOS ONE